# Assessing the Pathogenicity of Two Bacteria Isolated from the Entomopathogenic Nematode *Heterorhabditis indica* against *Galleria mellonella* and Some Pest Insects

**DOI:** 10.3390/insects10030083

**Published:** 2019-03-26

**Authors:** Rosalba Salgado-Morales, Fernando Martínez-Ocampo, Verónica Obregón-Barboza, Kathia Vilchis-Martínez, Alfredo Jiménez-Pérez, Edgar Dantán-González

**Affiliations:** 1Doctorado en Ciencias, Instituto de Investigación en Ciencias Básicas y Aplicadas, Universidad Autónoma del Estado de Morelos, Av. Universidad 1001, Chamilpa, 62209 Cuernavaca, Morelos, Mexico; salgadomoralesr@hotmail.com; 2Laboratorio de Estudios Ecogenómicos, Centro de Investigación en Biotecnología, Universidad Autónoma del Estado de Morelos, Av. Universidad 1001, Chamilpa, 62209 Cuernavaca, Morelos, Mexico; fernando.martinezo@uaem.mx (F.M.-O.); veronica.obregon@uaem.mx (V.O.-B.); 3Centro de Desarrollo de Productos Bióticos, Instituto Politécnico Nacional, Calle Ceprobi No. 8, San Isidro, Yautepec, 62739 Morelos, Mexico; vilchisk78@gmail.com (K.V.-M.); aljimenez@ipn.mx (A.J.-P.)

**Keywords:** entomopathogenic nematode, *Heterorhabditis indica*, bacteria, pathogenicity, insects

## Abstract

The entomopathogenic nematodes *Heterorhabditis* are parasites of insects and are associated with mutualist symbiosis enterobacteria of the genus *Photorhabdus*; these bacteria are lethal to their host insects. *Heterorhabditis indica* MOR03 was isolated from sugarcane soil in Morelos state, Mexico. The molecular identification of the nematode was confirmed using sequences of the ITS1-5.8S-ITS2 region and the D2/D3 expansion segment of the 28S rRNA gene. In addition, two bacteria HIM3 and NA04 strains were isolated from the entomopathogenic nematode. The genomes of both bacteria were sequenced and assembled de novo. Phylogenetic analysis was confirmed by concatenated gene sequence datasets as *Photorhabdus luminescens* HIM3 (16S rRNA, 23S rRNA, *dnaN*, *gyrA*, and *gyrB* genes) and *Pseudomonas aeruginosa* NA04 (16S rRNA, 23S rRNA and *gyrB* genes). *H. indica* MOR03 infects *Galleria mellonella*, *Tenebrio molitor*, *Heliothis subflexa*, and *Diatraea magnifactella* larvae with LC50 values of 1.4, 23.5, 13.7, and 21.7 IJs/cm^2^, respectively, at 48 h. These bacteria are pathogenic to various insects and have high injectable insecticide activity at 24 h.

## 1. Introduction

The entomopathogenic nematodes (Rhabditida: Heterorhabditidae and Steinernematidae) are parasites of insects that are associated in mutualist symbiosis with enterobacteria of the genera *Photorhabdus* and *Xenorhabdus* [1,2]. The entomopathogenic nematodes, unlike other parasites, cause the death of the host insect in a short time (within 24–48 h of infection), and this lethality is provided by associated bacteria [2].

The infective juvenile stage (IJ) is the only free life form of the entomopathogenic nematodes and functions as a vector carrying the symbiotic bacteria in the intestine to release them in the host insect’s hemolymph [3]. The life cycle of the nematode is completed inside the insect cadaver.

The symbiotic bacteria of entomopathogenic nematodes (*Photorhabdus* and *Xenorhabdus*) are gammaproteobacteria, Gram-negative, and members of the family Enterobacteriaceae. Canonically, *Photorhabdus* and *Xenorhabdus* establish monoxenic associations with *Heterorhabditis* and *Steinernema*, respectively. They are virulent to a wide variety of insects including Coleoptera, Hemiptera, Lepidoptera, and Diptera. The mean lethal dose (LD50) is low: <100 colony forming unit (CFU) injected into the hemolymph of *Galleria mellonella* larvae L (Lepidoptera: Pyralidae) [4,5,6,7,8]. The diversity of the toxins produced, lipasses, and proteases, as well as the ability of these bacteria to suppress the insect’s immune system, contribute to its high virulence. In addition, the secondary metabolites produced by the bacteria are essential for the development of the nematode [9].

However, non-canonical bacteria such as *Providencia* sp., *Ochrobactrum* sp., *Pseudomonas* sp., and *Alcaligenes faecalis* have been isolated from different entomopathogenic nematodes, and few studies have focused on the role of these bacteria and their association with the nematode [10,11,12,13]. Due to the lifestyle of entomopathogenic nematodes, it is interesting to evaluate the pathogenic activity of non-canonical bacteria in insects. Particularly, *Heterorhabditis indica* has been reported in diaxenic association with *Photorhabdus luminescens* and *Ochrobactrum* spp., and the latter does not have insecticidal activity against *G. mellonella.* In contrast, *A. faecalis* from *Heterorhabditis* sp. is pathogenic to *G. mellonella*, suggesting a possible role in pathogenesis in insects [11,12,13].

*G. mellonella* is widely used as a model insect to evaluate bacterial pathogenesis and virulence [14,15]. The larvae of *G. mellonella* have technical advantages such as that they are easy to maintain under laboratory conditions, no special equipment is needed, and virulence bioassays by injection can be made using larvae of the last instar. Subsequently, bacterial virulence can be determined by calculating the LD50 [16,17]. Particularly, *G. mellonella* has been used extensively for the study of the pathogenicity of entomopathogenic nematodes and their symbiotic bacteria [18]. However, there is great interest in evaluating the pathogenicity of entomopathogenic nematodes and their symbiotic bacteria (canonical and non-canonical) in other insects, mainly in insects considered pests of agricultural crops.

In this study, we report the isolation, identification, and pathogenicity of the entomopathogenic nematode *Heterorhabditis indica* MOR03 and its associated bacteria (canonical and non-canonical) in larvae of the *G. mellonella* model and larvae of pest insects: *Tenebrio molitor* L. (Coleoptera: Tenebrionidae), *Heliothis subflexa* Dyar (Lepidoptera: Noctuidae), and *Diatraea magnifactella* Dyar (Lepidoptera: Crambidae).

## 2. Materials and Methods

### 2.1. Isolation of the Entomopathogenic Nematodes

A total of 23 soil samples were collected in agricultural soils with a history of pesticide application of Morelos, Mexico, in the months of February to June 2014. Approximately 1 kg of soil was taken at a depth of 10 to 30 cm and the isolation of entomopathogenic nematodes was carried out using *G. mellonella* last-instar larvae as bait [19]. In the laboratory, the larvae were monitored daily, and the dead larvae were removed every day. Dead larvae with symptoms of infection by entomopatogenic nematodes such as absence of odor, absence of contaminating organisms, and coloration were recovered and disinfected with 5% sodium hypochlorite and rinsed 3 times with sterile distilled water before being transferred to White traps where nematodes were obtained from the insect larvae [20]. Koch postulates were performed to confirm that the nematodes were the cause of death [21].

### 2.2. Maintenance of Insects and Nematodes

The insects (*G. mellonella*, *H. subflexa*, and *D. magnifactella*) were kept at a temperature of 27 ± 2 °C with a light/darkness (LD) photoperiod of 12:12 h, kept at relative humidity (RH) 70 ± 10%, and grown on artificial diets. Only *T. molitor* was maintained at room temperature. The nematodes were reproduced into last-instar *G. mellonella* larvae and the IJs were recovered from the White traps after 12–14 days of inoculation with IJs.

### 2.3. Molecular Identification of the Entomopathogenic Nematodes

Genomic DNA (gDNA) was extracted from IJs using the DNA Isolation Kit for Cells and Tissues (Roche, Indianapolis, IN, USA, following the manufacturer’s instructions). The primers TW81 and AB28 were used for amplification of the DNA fragment containing the ITS1-5.8S-ITS2 region of rRNA genes [22]. The D2/D3 expansion segment of the 28S rRNA gene was amplified using the primers D2F and 536 [23,24]. PCR products were purified using a QIAquick Gel Extraction kit^®^ (Qiagen Inc., Santa Clara, CA, USA) and sequenced in both directions using the same primers at the Unidad de Síntesis y Secuenciación de DNA (USSD) of the Instituto de Biotecnología (IBT)-UNAM. The sequence reads were assembled and edited using BioEdit Version 7.2.5 software [25]. The resulting sequences were aligned using the Nucleotide BLAST tool NCBI GenBank (https://blast.ncbi.nlm.nih.gov/Blast.cgi). For phylogenetic reconstruction, we downloaded 25 ITS1-5.8S-ITS2 sequences and 19 D2/D3 sequences of the expansion segment of the 28S rRNA of species in the genera *Heterorhabditis* and *Caenorhabditis*. Multiple alignments between generated sequences and downloaded sequences selected from the GenBank database were made using the MUSCLE (version 3.8.31) program [26]. The jModelTest (version 2.1.10) [27] was used to select the model of nucleotide substitution using the Akaike information criterion (AIC). Phylogenetic trees were inferred based on maximum likelihood for both datasets using the PhyML (version 3.1) program [28] with 1000 bootstrap replicates. Phylogenetic trees were visualized and edited using the FigTree (version 1.4.3) program (http://tree.bio.ed.ac.uk/software/figtree/).

### 2.4. Determination of the LC50 of the Nematode against Insects

Bioassays to determine the mean lethal concentration, LC50, in *G. mellonella* and *T. molitor* consisted of using 10 and 15 larvae per dose tested, which were confined in 9-cm Petri dishes floored with Whatman no. 2 filter paper disks. Each treatment had five doses: 0.7, 1.5, 3.1, 6.2, 12.5, and 15.7 and 7.8, 15.7, 31.4, 62.8, and 157.2 IJs/cm^2^, respectively. The doses used to determine the LC50 for *H. subflexa* and *D. magnifactella* were 4, 8, 16, 32, and 64 and 4, 8, 16, 24, and 48 IJs/cm^2^, respectively. These last ones were confined individually in a 6-well plate Costar (Corning Life, New York, NY, USA). The concentrations of nematodes were adjusted by volumetric dilutions in a final volume of 1 mL according to the methodology described by Glazer and Lewis, 2000 [29]. In all cases, sterile distilled water was used as a negative control and larvae were incubated at 28 ± 2 °C. Mortality was evaluated 48 h after inoculation and at least three independent bioassays were conducted. Mortality data was analyzed using Probit (version 1.0) to determine the LC50 with PoloPlus software [30].

### 2.5. Bacteria Isolation from the Nematodes

Bacteria were isolated from the hemolymph of infected larvae with isolate *H. indica* MOR03. Disinfection of the larvae of *G. mellonella* was performed using 2% (*w*/*v*) benzalkonium chloride and rinsing with sterile distilled water before allocation into 9-cm Petri dishes. Subsequently, 1.5 IJs/cm^2^ (previously disinfected) were added. After 38 h of infection, the larvae were disinfected under sterile conditions and the hemolymph was obtained using insulin syringe (BD Medical-Diabetes Care, Holdrege, NE, USA) with a 31-gauge needle; the hemolymph droplet was inoculated on the nutrient bromothymol blue agar (NBTA) and incubated at 28 ± 2 °C for 72 h. To obtain a fresh culture of bacteria from the IJs, approximately 1000 IJs were disinfected with 0.1% (*w*/*v*) benzalkonium chloride solution for 20 min, then rinsed with sterile distilled water and macerated with a Teflon micropestle under sterile conditions. A 10-μL sample of the macerated material was inoculated in Luria–Bertani (LB) broth and NBTA medium and incubated at 28 ± 2 °C for 24 h [12]. The experiment was carried out three times using different generations of IJs of *H. indica* MOR03 and the identification of the bacteria was confirmed by sequencing of the 16S rRNA gene.

### 2.6. Sequencing, Assembly, and Genome Annotation

The gDNA of HIM3 and NA04 strains was extracted using the ZR Fungal/Bacterial Kit MiniPrep^TM^ (Zymo Research, Irvine, CA, USA, following the manufacturer’s instructions), and 5 μg of gDNA was sequenced with the Illumina HiSeq platform (2 × 300 bp Paired-End reads). Bacterial genomes were assembled de novo using the SPAdes (version 3.5.0) program [31] and automated annotation of bacterial genomes was performed using the RAST (version 2.0) server [32]. The annotations resulting from the genomes were used to search for virulence factors based on keywords (pathogenesis, virulence, toxin, adhesion, fimbriae). The prediction of rRNA and tRNA genes was performed using the RNAmmer (version 1.2) [33] and ARAGORN [34] servers, respectively [35,36].

### 2.7. Molecular Identification of Bacteria

The 16S rRNA, 23S rRNA, *dnaN*, *gyrA*, and *gyrB* sequences, and 16S rRNA, 23S rRNA, and *gyrB* sequences of housekeeping genes were obtained from the genomes of HIM3 and NA04 strains, respectively. The gene sequence datasets from the HIM3 and NA04 genomes were compared to six downloaded gene sequence datasets of the genera *Photorhabdus* and *Proteus*, and 16 downloaded gene sequence datasets of the genera *Pseudomonas* and *Escherichia*, which were obtained from the GenBank database (https://www.ncbi.nlm.nih.gov/) using the nucleotide BLAST suite (https://blast.ncbi.nlm.nih.gov/Blast.cgi). Multiple alignments between gene sequence datasets of the HIM3 and NA04 strains and downloaded gene sequence datasets selected from the GenBank database were made using the MUSCLE (version 3.8.31) program [26]. Alignment sequences per genome were concatenated using a Python script. The jModelTest (version 2.1.10) [27] was used to select the model of nucleotide substitution using the Akaike information criterion (AIC). Phylogenetic trees were inferred based on maximum likelihood for each concatenated gene dataset using the PhyML (version 3.1) program [28] with 1000 bootstrap replicates. Phylogenetic trees were visualized and edited using the FigTree (version 1.4.3) program (http://tree.bio.ed.ac.uk/software/figtree/).

### 2.8. Bacterial Pathogenicity against Insects

The bacterial isolates of the hemolymph (HIM3), the macerate of IJs (NA04), and *E. coli* DH5α (negative control) were individually inoculated in 100 mL of LB liquid medium and grown overnight at 30 °C and 150 rpm. Viable counts were made, and the doses were adjusted in a final volume of 10 μL. Each bacterial suspension was injected individually into an insect’s hemolymph at the last left pro-leg into *G. mellonella*, *T. molitor*, and *D. magnifactella* larvae by using insulin syringe (BD Medical-Diabetes Care, Holdrege, NE, USA) with a 31-gauge needle. LB medium, puncture, and equivalent doses of *E. coli* DH5α were used as negative controls. Approximately, the doses used on *G. mellonella* were 10–120 CFU HIM3 and 25–400 CFU NA04, and for *T. molitor*, 1000–10,000 CFU HIM3 and 400–5000 CFU NA04 were used. Mortality was determined 24 h after injection. *D. magnifactella* mortality was evaluated 24 and 36 h after injection at 3000 and 10,000 CFU of HIM3 and NA04, respectively. Fifteen larvae were used per dose and each larva was placed in a 9-cm Petri dish with 0.5 g of artificial diet and incubated at 28 ± 2 °C. The experiment was repeated in three independent bioassays and the data were analyzed using Probit version 1.0 to determine the LD50 with PoloPlus software [30].

## 3. Results

### 3.1. Isolation and Molecular Identification of H. indica

A single isolate of entomopathogenic nematode was obtained from one sample, corresponding to 4.3% of 23 samples. The positive sample was collected in a field cultivated with sugarcane in the town of Oacalco in Yautepec, Morelos, Mexico (18°55′16.9″ N and 99°02′24.0″ W). The infection characteristics present in the larvae of *G. mellonella* infected with entomopathogenic nematode were no odor or copper red coloration. The IJs emerged from the infected larvae between 12 and 14 days after infection.

The ITS1-5.8S-ITS2 sequence of our isolate MOR03 is 99% similar to *H. indica* (strain D1, GenBank accession number AY170329). Additionally, the 28S D2/D3 expansion segment sequence is 99% similar to *H. indica* (strain Cohen 21, GenBank accession number GU177842). Two maximum likelihood trees (Figure 1a,b) show that the entomopathogenic nematode isolated belongs to the *H. indica* species. The model of nucleotide substitution (selected by jModelTest) was TMV+G, for both ITS1-5.8S-ITS2 and 28S D2/D3 datasets.

### 3.2. Pathogenicity and Virulence of H. indica MOR03 to Insects

The isolated *H. indica* MOR03 is a pathogen to several Lepidoptera—*G. mellonella*, *H. subflexa*, and *D. magnifactella*—and the coleopteran *T. molitor* in laboratory conditions. The values of LC50 show that the order of virulence of *H. indica* MOR03 is *G. mellonella* > *H. subflexa* > *D. magnifactella* > *T. molitor* (Table 1). The dose–response curves are shown in Appendix A.

### 3.3. Isolation of Bacteria, Genome Sequencing, and Molecular Identification

Two bacterial isolates were obtained; HIM3 was isolated from the hemolymph of larvae infected with *H. indica* MOR03; HIM3 and NA04 were isolated from the maceration of IJs in different generations. In addition, the bacterial genomes of the HIM3 and NA04 strains were sequenced and assembled de novo. The draft genomes of HIM3 and NA04 strains have a total length of 5,472,253 bp in 91 contigs with a G+C content of 42% and a total length of 6,375,895 bp in 54 contigs with a G+C content of 66.4%, respectively. Both genome sequences were deposited in the GenBank database under accession numbers GCA_002204205.1 and GCA_002204155.1, respectively [35,36]. The automated annotation of the bacterial genomes shows that the HIM3 strain contains 4659 coding sequences (CDS), 70 tRNAs, and 10 rRNAs; the NA04 strain contains 5823 CDS, 67 tRNAs, and 7 rRNAs. Maximum likelihood trees of five (16S rRNA and 23S rRNA, *dnaN*, *gyrA*, and *gyrB*) and three (16S rRNA, 23S rRNA, and *gyrB*) concatenated gene sequence datasets show that the HIM3 and NA04 strains are grouped with the *Photorhabdus luminescens* and *Pseudomonas aeruginosa* species, respectively (Figure 2a,b). The HIM3 strain is closely related to *Photorhabdus luminescens* DSM 3368 (GenBank accession number NZ_JXSK00000000.1) isolated from *H. bacteriophora* in Australia; the NA04 strain is related to *Pseudomonas aeruginosa* M18 (GenBank accession number CP002496) isolated from rhizosphere soil of sweet melon [37]. Therefore, we named these bacteria *Photorhabdus luminescens* HIM3 and *Pseudomonas aeruginosa* NA04.

The genome of *P. luminescens* HIM3 contains CDS that code for proteins involved in pathogenesis, such as effector proteins (VirK and YopT), lipoproteins involved in adhesion and colonization (NlpE, ShlA, galactophilic lectin PA-I), fimbrial adhesion precursors, siderophore production and iron-regulated proteins, proteins involved in the evasion of the immune system (palmitoyltransferase pagP), hemolysins, and a large number of insecticidal toxins. On the other hand, the genome of *P. aeruginosa* NA04 effector proteins (YopR, HopPmaj) includes exotoxin A precursor, accessory cholera enterotoxin, and zona occludens toxin, as well as hemolysin, adhesion, and siderophore production proteins. Both HIM3 and NA04 strains contain genes that code for toxins of the RTX family (Appendix A).

### 3.4. Pathogenicity of P. luminescens HIM3 and P. aeruginosa NA04 in Insects

The bacteria *P. luminescens* HIM3 and *P. aeruginosa* NA04 are pathogenic to the larvae of *G. mellonella* and cause a high percentage of mortality at 24 h. The LD50 value of *P. luminescens* HIM3 to *G. mellonella* is 19 (14.4–23.8 95% CI) CFU/larvae, χ^2^ = 16.1. Similarly, when 25 to 400 CFU of *P. aeruginosa* NA04 were injected into *G. mellonella* larvae, almost 100% mortality rates were observed (Figure 3). These results indicate that both HIM3 and NA04 isolates are highly virulent to *G. mellonella*. The bioassays with *T. molitor* showed that a high dose (1491 CFU/larva, *χ*^2^ = 7.9) of *P. aeruginosa* NA04 is required to cause 50% mortality at 24 h and that doses (1000–10,000 CFU) of *P. luminescens* HIM3 cause between 20% and 40% mortality in *T. molitor* at 24 h. Interestingly, no mortality was observed at 24 h when 3000 and 10,000 CFU of HIM3 and NA04 were injected into the hemolymph of *D. magnifactella*. However, at 36 h, approximately 100% mortality was observed with HIM3 and only 10% mortality with NA04 (Figure 4).

## 4. Discussion

*Heterorhabditis indica*’s preference is for regions with warm climates, including tropical and subtropical regions in North and South America, the Caribbean, Africa, Asia, and Australia [38]. The entomophatogenic nematode *H. indica* MOR03 was isolated from agricultural soil cultivated with sugarcane in the town of Oacalco in Yautepec, Morelos, where the average temperature is around 22.7 °C and the climate is warm and subhumid. *H. indica*’s preference for warm climates suggests that it possesses advantageous characteristics to inhabit this geographic zone; this may include adaptation to high temperatures, tolerance to desiccation, and good dispersibility [39].

The main phylogenetic analyses of the genus *Heterorhabditis* are based on the partial 18S, 28S, ITS1, and ITS1-5.8-ITS2 regions of rRNAs genes. Nevertheless, the analysis of the D2/D3 expansion segment of the 28S rRNA gene resolved the relationship between *Heterorhabditis* species. Actually, only one genome of genus *Heterorhabditis* has been reported in the GenBank database: the draft genome of *H. bacteriophora* M31e (GenBank accession number ACKM00000000.1). Therefore, there is not enough information available to perform a phylogenetic analysis of concatenated housekeeping genes or the core genome. In this work, we selected two different regions to identify the species of the entomopathogenic nematode isolated: ITS1-5.8S-ITS2 and the D2/D3 expansion segment of rRNA genes. Also, we used a character-based method for phylogenetic reconstruction, the maximum likelihood (ML) method, which is a probabilistic method and requires the selection of a model of nucleotide substitution from 88 models available. We highlight the importance of sequencing the genomes of the genus *Heterorhabditis* to reliably study the evolutionary history, geographical distribution and ecology of this genus.

*H. indica* MOR03 is a pathogen in all tested insects but with different degrees of virulence depending on the host insect. According to other reports, the LC50 varies with respect to the nematode species, its virulence, and the host insect. For example, [40] reported that 25 (10.2 IJs/cm^2^) and 55 IJs (20.5 IJs/cm^2^) are required to achieve 50% mortality of *G. mellonella* with *H. indica*, but almost three times more *H. bacteriophora*, 160 IJs, (66.6 IJs/cm^2^) are needed to achieve the same mortality. Our data show that *H. indica* MOR03 is more virulent than this, as only 1.4 IJs/cm^2^ were required to kill 50% of the *G. mellonela* larvae. A similar LC50, 1.99–6.9 IJs per *G. mellonella* larva (0.9–3.4 IJs/cm^2^), was reported for the four isolates of *H. indica* [18].

The values of LC50 that we obtained for *T. molitor*, *H. subflexa*, and *D. magnifactella* are 9 to 16 times larger than that for *G. mellonella*. This confirms that *G. mellonella* is the most susceptible host insect to *H. indica* MOR03 and is consistent with other reports [18]. The LC50 values of *H. indica* MOR03 on insects of economic interest as *H. subflexa* and *D. magnifactella* (<25 IJs/cm^2^) show its potential as a biological control agent [41]. However, field studies are required for validation.

In this work, we evaluated the pathogenicity and virulence of two bacteria isolated from *H. indica* MOR03 against insects. An interesting aspect is that, in addition to *P. luminescens* HIM3, the non-canonical bacterium *P. aeruginosa* NA04 exhibited entomopathogenic activity in some insects, but not in *D. magnifactella*. Both bacteria are pathogens and highly virulent (almost 100% mortality after 24 h) against *G. mellonella*. A 100% mortality rate after 24 h was reported when injecting the same dose of *P. aeruginosa* CFU into *G. mellonella* [42]. However, susceptibility to HIM3 and NA04 is affected by the host insect. *T. molitor* required doses in excess of 1000 CFU. However, *D. magnifactella* was affected by *P. luminescens* HIM3 and not by *P. aeruginosa* NA04. This suggests that *P. luminescens* HIM3 has a larger arsenal of insecticidal toxins or greater specificity for *D. magnifactella.*

Interestingly, the non-canonical bacterium *P. aeruginosa* NA04 was not found in the hemolymph of *G. mellonella* larvae infected with *H. indica* MOR03 at 36 h after infection. However, it is possible that the non-canonical bacterium *P. aeruginosa* NA04 is released or reproduces in the hemolymph in later hours. Furthermore, the presence of non-canonical bacteria in JIs from different generations suggests that the bacteria are retained by *H. indica* MOR03.

The non-canonical bacteria *P. aeruginosa* NA04 could be contributing to the pathogenesis of the nematode only in some insects (*G. mellonella* and *T. mollitor*) in laboratory conditions. In contrast, *P. luminescens* HIM3 is pathogenic for all test insects. Both bacteria contain in their genome the genes that code for the lectins involved in the process of colonization and invasion of the host, such as galactophilic lectin PA-I and proteins for the synthesis of fimbria, another important factor for colonization [43,44]. The genome of *P. luminescens* HIM3 presents the gene coding for a palmitoyltransferase (pagP), which modifies the lipid A of lipopolysaccharide (LPS); this modification allows the bacteria to inhibit the inflammatory response and evade the insect immune response [45]. The PhoP–PhoQ two-component system is associated with virulence in *P. luminescens* and *P. aeruginosa*, and it is contained in the genome of HIM3 and NA04 [46,47,48]. The acquisition of iron in pathogenic bacteria is essential to overcome the immune response of the host. The strains HIM3 and NA04 have a variety of genes for the production of siderophores, small iron binding molecules that are secreted by bacteria to acquire ferric iron the host [49]. In particular, *P. luminescens* HIM3 contains large proteins with possible insecticidal activity as well as proteins that frame the type III secretion system (T3SS) responsible for introducing the virulence proteins into the cytoplasm of eukaryotic cells; the differences in the range of effector proteins secreted by each bacterium could be related to its degree of virulence and specificity [50]. In addition to pathogenicity, T3SS plays an important role in symbiotic relationships [51,52]. It should be noted that the presence of the nematode plays an important role in the suppression of the host´s immune system, as the *H. bacteriophora* cuticle causes host immunosuppression through the inhibition of eicosanoid biosynthesis in *G. mellonella* [53].

## 5. Conclusions

This paper reports the isolation and molecular identification of the nematode *H. indica* MOR03 that has proven to be lethal for various insects. The insecticidal activity could be related to the presence of two bacteria (canonical and non-canonical) which were also identified. The relationship of the canonical bacteria of the genus *Photorhabdus* with entomopathogenic nematodes of the genus *Heterorhabditis* has been extensively reported. However, there have been few studies focused on the role played by non-canonical bacteria. This report shows that *P. aeruginosa* NA04 has insecticidal activity for different insects but that this activity is different from that of *P. luminescens* HIM3, which would help to clarify how symbiotic bacteria work when the nematode infects different hosts. They would also provide new resources and alternatives for further studies of symbiotic bacteria of the entomopathogenic nematode, for example, the role that bacteria play in the specificity that the nematode has on insects, due to toxins and other mechanisms that contain the associated bacteria. Entomopathogenic nematodes and their associated bacteria (canonical and non-canonical) are an appropriate model to test hypotheses regarding the role that each organism plays in the various activities of the nematode and to understand the complex mechanisms of symbiotic relationships.

## Figures and Tables

**Figure 1 insects-10-00083-f001:**
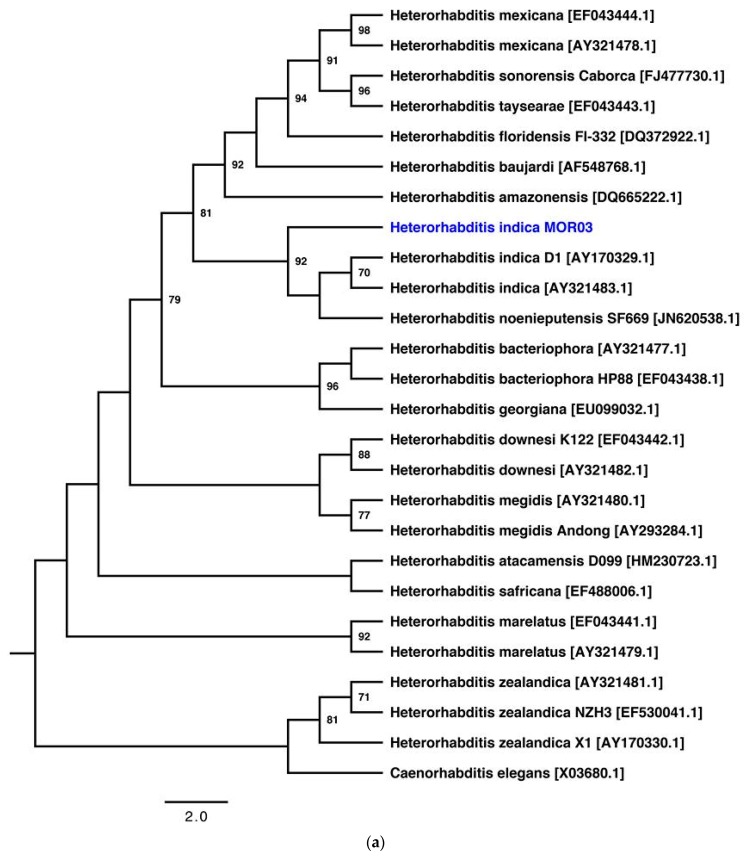
Phylogenetic relationships based on the regions of rRNA gene sequences of *Heterorhabditis indica* MOR03 (blue letters) with 24 and 17 sequences of genus *Heterorhabditis*, respectively. (**a**) Phylogenetic tree of the ITS1-5.8S-ITS2 region of rRNA gene sequences. (**b**) Phylogenetic tree of the D2/D3 expansion segment of 28S rRNA gene sequences. Phylogenetic trees were obtained using the PhyML (version 3.1) program with the maximum likelihood method and 1000 bootstrap replicates. Bootstrap values are displayed as a percentage of confidence (>70%) in the nodes.

**Figure 2 insects-10-00083-f002:**
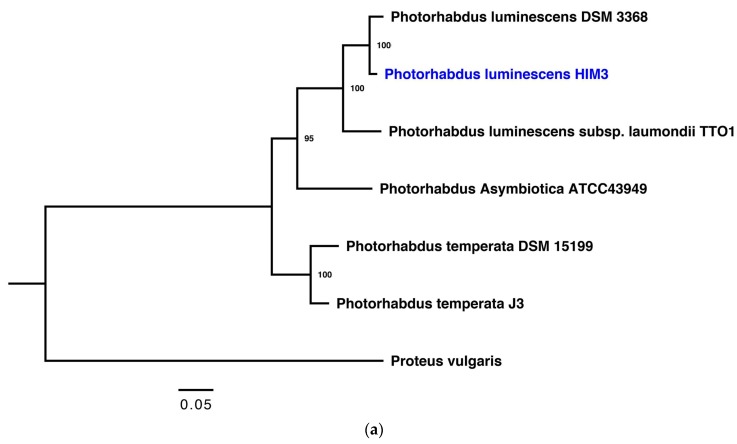
Phylogenetic relationships based on the concatenated gene sequence datasets of the two bacteria strains HIM3 and NA04 isolated from the entomopathogenic nematode *H. indica* str. MOR03. (**a**) Phylogenetic tree based on the concatenated sequences of five housekeeping genes (16S rRNA and 23S rRNA, *dnaN*, *gyrA*, and *gyrB*) of *Photorhabdus luminescens* str. HIM3 (blue letters) with five concatenated gene sequence datasets of genus *Photorhabdus*. (**b**) Phylogenetic tree based on concatenated sequences of three housekeeping genes (16S rRNA, 23S rRNA, and *gyrB*) of *Pseudomonas aeruginosa* str. NA04 (blue letters) with 15 concatenated gene sequence datasets of genus *Pseudomonas.* Phylogenetic trees were obtained using the PhyML (version 3.1) program with the maximum likelihood method and 1000 bootstrap replicates. Bootstrap values are displayed as a percentage of confidence (>70%) in the nodes.

**Figure 3 insects-10-00083-f003:**
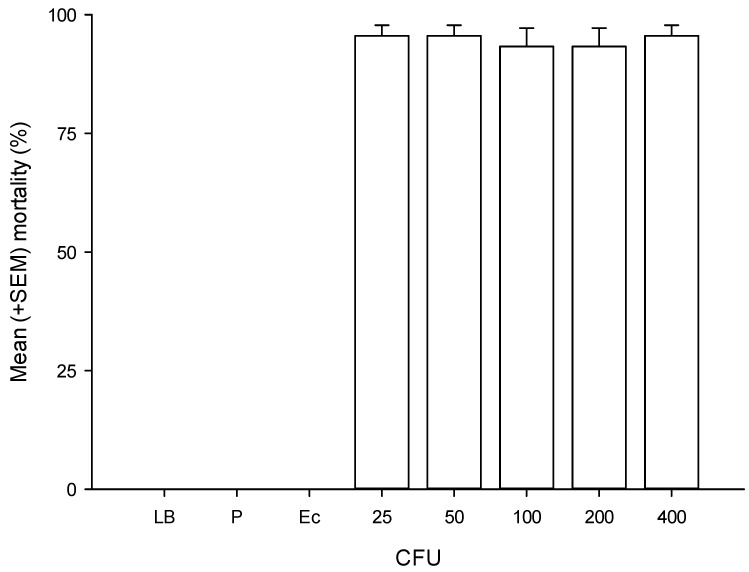
Mean (± SEM) mortality of *Galleria mellonella* after inoculation with *P. aeruginosa* NA04. LB = Luria–Bertani broth, P = puncture, Ec = *E. coli* DH5α (25, 50, 100, 200, and 400 CFU).

**Figure 4 insects-10-00083-f004:**
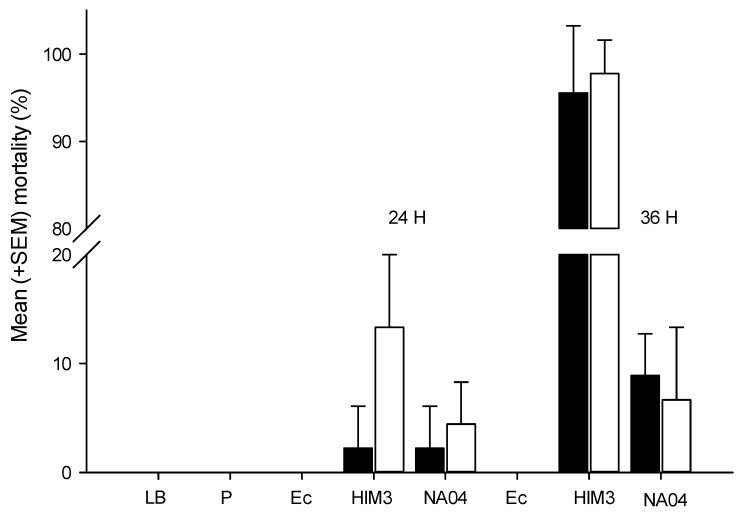
Mean (± SEM) mortality of *D. magnifactella* at 24 and 36 h after inoculation with HIM3 and NA04 at 3000 (empty bars) and 10,000 CFU (solid bars), respectively. Error bars are the SEM. LB = Luria–Bertani broth, P = puncture, Ec = *Eschericia coli* (3000 and 10,000 CFU) HIM3 = *P. luminescens* HIM3, NA04 = *P. aeruginosa* NA04.

**Table 1 insects-10-00083-t001:** LC50 (IJs/cm^2^) values of *Heterorhabditis indica* MOR03 against *G. mellonella*, *T. molitor*, *H. subflexa*, and *D. magnifactella* obtained at 48 h.

Insect	*n*	Instar	LC50 (IJs/cm^2^) 95% CI	χ^2^	*df*
***G. mellonella***	320	Last instar	1.4 (1.0–1.9)	33.8	30
***T. molitor***	210	15th instar	23.5 (15.8–33.3)	6.9	12
***H. subflexa***	225	Last instar	13.7 (10.1–18.3)	12.3	13
***D. magnifactella***	204	5th instar	21.7 (14.4–43.8)	7.7	15

Larvae were incubated at 28 ± 2 °C in darkness. (*p* < 0.05).

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
