# Peer review of "Assessing the Pathogenicity of Two Bacteria Isolated from the Entomopathogenic Nematode Heterorhabditis indica against Galleria mellonella and Some Pest Insects"

_insects, 2019, doi:10.3390/insects10030083_

Round 1

Reviewer 1 Report

The paper present interesting data properly analysed; just two notes:

- about the scientific names: pay attention to the Authorities when they are reported for the first time in the text

- the conclusions needs to be improved

Author Response

We appreciate your comments and suggestions. We send a letter answering your comments point by point. We hope our response is satisfactory.

Reviewer 2 Report

All the comments to improve the manuscript  are shown in the attached document

Author Response

(The authors gave the same response as above.)

Reviewer 3 Report

Dear authors,

Please do not get discouraged by the many comments. I think this is a really interesting paper, which however severely suffers from poor structuring and writing style. In an improved form, this will be a really nice paper, which I will be happy to cite myself in future work. (I am myself working on investigating the pathogenicity of a to-be described yeast using the Galleria mellonella model.)

I look forward to a revised version of this manuscript.

GENERAL COMMENT

Please have an English native speaker read and edit the revised version of the manuscript. There are quite many mistakes that should be corrected before publication.

TITLE

Lines 2-5: Please revise the title. The current title is misleading. Perhaps a broader more general title would be better here. What is the purpose of this paper? “Evaluating a nematode-bacteria system using phylogenetic and Galleria model pathogenicity”. Something like this should be the title. Informative, without too much detail (e.g. please do not include isolate information “MOR03” in the title).

ABSTRACT

Line 23: “D2/D3 expansion segment of the 28S rRNA gene.”

INTRODUCTION

General comment: I like an introduction to flow nicely, starting very broad, then funneling down to the study organisms at hand. Here, the introduction is quite confusing. Starting very detailed with nematodes, then moving to their bacteria, something about pathogenicity (without properly introducing the Galleria mellonella pathogenicity model!!), some specifics about a single species of nematode but without it being made clear why, ending with a general statement that studying the nematode-bacteria complex is important to unravel the nature of the nematode-bacteria interaction.

My suggestion is to revise the entire introduction. Start the first paragraph with what are entomopathogenic nematodes, why they are important to study, (e.g. see Haelewaters et al. 2017, citation below), etc. Then (perhaps the next paragraph depending on how much you write about the nematodes), introduce the symbiotic bacteria (= funneling down). In the following paragraph, introduce the Galleria pathogenicity model and make a link to the nematode-bacteria system. End the introduction with what this paper will add to what is already known about these nematode-bacteria interactions (= ending broad again, nicely flowing into the methods). Currently, it is unclear to me what the purpose of this study is. That should be brought forward in the final paragraph of the introduction.

Line 35: A good citation to include about entomopathogenic nematodes is the recent review by Haelewaters et al. (2017). Parasites of Harmonia axyridis: current research and perspectives. BioControl Volume 62, Issue 3, pp 355–371.

Line 44: “… an entomopathogenic bacterium …” (singular form)

Line 45: this is the first time Galleria mellonella is mentioned in the manuscript. Write the genus name in full. Please introduce what this is.

Line 50: At this point in the manuscript, several genera starting with the letter P have been introduced, so please write the name of this bacterium in full here. I assume this is “Photorhabdus luminescens”? (also correct the typo in the epithet!)

METHODS

Lines 61-69: Please add when these soil samples were collected. Were there specific time ranges? Random times for collections?

Line 67: “… before being transferred …”

Line 68: What are Koch pustulates? There should be at least a reference here, or some sort of short explanation for readers who do not study microbial pathogenicity.

Lines 70-76 (and throughout the manuscript): Do not start sentences with abbreviations. “Galleria mellonella larvae were fed …” “Tenebrio molitor larvae were reared …” Etc.

Also, please add systematic classification to each newly introduced taxon in the manuscript. It makes no sense to have all these names if the reader doesn’t know what organisms they are. E.g. “Tenebrio molitor larvae (Coleoptera, Tenebrionidae) were reared …”

Line 83: Please add whether (or not) the same primers were used for sequencing. “PCR products were sequenced using the same primers at the …”

Lines 83-87: Please replace “PCR sequences” by “generated sequences”. Additionally, forward and reverse sequence reads must be assembled in some sort of a program, such as Sequencher before comparisons can be made. Please add these steps. Finally, the sentence is too long. Line 84 states that the sequences were compared but only in line 87 the tool for comparison is mentioned. Split this sentence up in at least two sentences.

Something like this: “Assembly and editing of sequence reads was done in Sequencher 4.10.1 (Gene Codes Corporation, Ann Arbor, Michigan). Resulting sequences were blasted using the Nucleotide BLAST tool from NCBI GenBank (https://blast.ncbi.nlm.nih.gov/Blast.cgi). For phylogenetic reconstruction, we downloaded 25 ITS1-5.8S-ITS2 sequences and 19 D2/D3 28S sequences of species in the genera Heterorhabditis and Caenorhabditis.”

Line 88: Again replace “PCR sequences” with “generated sequences”, and replace “know sequences selected from GenBank database” by “downloaded sequences”. As a general rule, a paper benefits from brevity. Please keep this in mind when revising.

Line 91: “… inferred for both datasets …”

Line 106: “… infected larvae with isolate H. indica MOR03.”

RESULTS

Line 158: just one entomopathogenic nematode or multiple ones?

Lines 164-169: Is it important to provide all this detail? You just gave your own isolate the label “MOR03”, that’s basically it. Given the 99% identity, is it then Heterorhabditis indica or something else very closely related? That is what I would like to see written here. In addition, % identity is found with BLAST searches. Phylogenetic trees provide relationships. The bootstrap support numbers then add value to these retrieved relationships. Please clarify your findings better given this distinction. The paragraph can be structured as follows: “Our sequence of isolate MOR03 is 99% identical to … In the phylogenetic reconstruction(s), it clusters with isolates of H. indica with high support (BS = XX).”

Comment on Figures 1a,b and 2a,b: Bootstrap values should be given as values between 1 and 100. Usually, only BS > 70 are provided. I do not know what the values in the shown trees are, but they are not bootstrap support values. Please revise.

DISCUSSION

Line 338: “Heterorhabditis indica’s preference for warm climates …”

Line 349: “confirming that the latest”? => please stop the sentence after “G. mellonella” and start a new sentence with: “This confirms that the latter (?) is …”

Lines 356, 362, and other places in the manuscript: the singular form of “species” is also “species”. Please correct.

Line 359: “… and requires the selection of a model of nucleotide substitution.”

Line 360: The TVM+G model was selected for both the ITS1-5.8S-ITS2 and D2/D3 28S datasets? This information should be provided in the Results section, not in the discussion. To me, lines 357 to 362 are unfitting in a Discussion.

CONCLUSIONS

Lines 384-393: Please revise the conclusion incorporating revisions that result from the above comments.

Author Response

(The authors gave the same response as above.)

Reviewer 4 Report

Review report of the manuscript entitled Pathogenicity against insect larvae of two bacteria isolated of new molecularly identified entomopathogenic nematode Heterorhabditis indica MOR03

The article describes the pathogenic activity of a nematode and associated bacteria to different laboratory insect models. The paper is interesting because it studies pathogenicity of the a complex nematode-bacteria. However, there are some points that I think would help to improve the manuscript.

1.       In first place, the introduction is too short and poorly introduces, both, pathogens and their hosts. Particularly last part of introduction fails to state clearly the scientific question or working hypothesis including the importance/application, biological control maybe? Some hypothesis or question could be related for example with cooperation between bacteria and nematode to kill the host or who is determining for the pathogenesis: bacteria or the nematode, both?

2.       The article does not discuss or try to identify the actual host of the nematode and the associated bacteria. Insect models such as Galleria and Tenebrio are useful to inform about pathogenicity but none of them are pests for sugar cane. It would be useful in the discussion to talk about this point.

3.       Beyond species identification, the phylogeny is merely presented and poorly discussed. For example, clustering informs about the ecology and evolution of the specie but this information is not exploited at all.

4.       P. luminescens is an insect pathogen associated with nematodes but this is not the case of P. aureuginosa. Pseudomonas is common in soil and many other environments. Could it be that the presence of this bacterium is incidental? It is actually known that P. aeruginosa is able also to kill nematodes and there is even a good stablished host-pathogen model with C. elegans. Is P. aeruginosa also pathogenic for Heterorhabditis indica?

5.       If both bacteria are inoculated together, which may be the case in nature scenarios, what outcome are authors expecting? Is it possible to co-cultivate these two species together without conflicts? 

6.       A technical question, as a microbiologist I´m intrigued about the ability of experimentalist to inoculate exactly 25, 50, 100 and so on CFUs. How the authors determine such numbers? Plating bacteria to determine CFU is quite variable. Even when researchers select for different bacterial doses it is better to use ´approximately´ term. In addition, the accurate enumeration of bacteria incline researcher to use doses that are related to each other by half or a log of difference. It is very common to find inoculation counts of 102, 103, 104 and so on to deal with this issue. By plating a culture, in some technical replicas of plates you might find 100 colonies or 200 within the same culture.   

7.       For survival analysis, Kaplan-Meier curves would be more appropriate because it is more informative and killing dynamic. However, the authors decided to determine mortality mostly after 24 hours. Is there any particular explanation?

8.       It is also surprising to see that mortality of puncture control is zero. It is common in Galleria to observe some mortality even in control groups. Unless that injection method is extremely refined.  

9.       The statistical analysis is not well described or absent.

10.   The table S1 and S2 should indicate references that authors used to retrieve the information about possible bacterial virulent factors for insect because they did not conduct any experiment to find out that information.    

Author Response

(The authors gave the same response as above.)

Round 2

Reviewer 2 Report

I send you my comments in attachment 

Author Response

We thank the reviewer for their interest in our manuscript, and for all of their helpful comments. Provided below is a point-by-point response describing our attempts to address, and when possible incorporate, all of their requested revisions in our manuscript.

Reviewer 3 Report

Dear Authors,

The manuscript is much improved compared to the previous version I have reviewed. I do have a few further suggestions. Please carefully revise Figures 1 (a & b) and 2 (a & b). 

Lines 110-113: Suggestion for improvement:

"jModelTest (version 2.1.10) [28] was used to select the model of nucleotide substitution using the Akaike information criterion (AIC). Phylogenetic trees were inferred based on maximum likelihood for both datasets using the PhyML (version 3.1) program [29] with 1000 bootstrap replicates."

=> Note how jModelTest should be written.

=> This means that the program does 1000 replicates. However, the bootstraps that are to be shown on the trees must be between 1 and 100. This is common practice. I have never seen bootstraps up to 1000 (because it is thought of the % for confidence of a given node, even though we don't write the % symbol). Usually, only bootstrap values that are informative are shown, = those values >70. (Because if you have a node that is supported by a bootstrap of, say, 45, then in only 450 of the replicate trees out of 1000 this node found. So a bootstrap of 45 gives no confidence for a given node.)

Please make appropriate changes to Figures 1a,b and 2a,b. Change all bootstrap values to numbers between 1 and 100, and only include bootstrap values >70. (You will see that the structure within H. indica collapses because of low bootstrap values, which is good -- they're all the same species.)

Line 159-162: Again, the following improvements are suggested:

"JmodelTest (version 2.1.10) [28] was used to select the model of nucleotide substitution using the Akaike information criterion (AIC). Phylogenetic trees were inferred based on maximum likelihood for each concatenated gene dataset using the PhyML (version 3.1) program [29] with 1000 bootstrap replicates.

Lines 187-192: I made a few changes in these sentences for clarity:

"The ITS1-5.8S-ITS2 sequence of our isolate MOR03 is 99% similar to H. indica (strain D1, GenBank accession number AY170329). Also the 28S D2/D3 sequence is 99% similar to H. indica (strain Cohen 21, GU177842). Two maximum likelihood trees (Fig. 1a and b) show that the entomopathogenic nematode isolated belongs to the H. indica species. The model of nucleotide substitution (selected by jModelTest) was TMV+G, for both ITS1-5.8S-ITS2 and 28S D2/D3 datasets."

Otherwise, the manuscript seems fine to me to be acceptance.

Very best wishes,

Danny

Author Response

Your comments were highly insightful and enabled us to greatly improve the quality of our manuscript. Thank you!

Reviewer 4 Report

The reviewed version of the article greatly improved the manuscript. My queries were fulfilled. 

Author Response

Thank you for your assessment.